# Attendee's awareness about preventive chemotherapy neglected tropical diseases (PC-NTD) control during the first world neglected tropical diseases day in Ekiti State, Nigeria

**Hammed O. Mogaji**[1]*, **Ikenna M. Odoh**[2], **Concilia I. Iyeh**[1], **Abdulhakeem A. Adeniran**[3], **Segun I. Oyedeji**[1], **Hilary I. Okoh**[1], **Adedotun A. Bayegun**[4], **Olaitan O. Omitola**[4], **Cynthia U. Umunnakwe**[4], **Francisca O. Olamiju**[5], **Olatunwa J. Olamiju**[5], **Uwem F. Ekpo**[4]

1 Department of Animal and Environmental Biology, Federal University, Oye-Ekiti, Ekiti, Nigeria, 2 University Medical Centre, Federal University, Oye-Ekiti, Ekiti, Nigeria, 3 Laboratory of Molecular Biomedicine, Centre for Genomic, Biotechnology, Instituto Politecnico Nacional, Reynosa, Tamaulipas, Mexico, 4 Department of Pure and Applied Zoology, Federal University of Agriculture, Abeokuta, Nigeria, 5 Mission To Save The Helpless (MITOSATH), Jos, Nigeria

* mogajihammed@gmail.com

## Abstract

### Background

The need to control Neglected Tropical Diseases (NTDs) and sustain progress towards elimination through mass administration of medicines requires substantial communal participation. This study, therefore, assessed the knowledge and perception of attendees' regarding NTDs and its control activities during the inaugural World NTD day event in Ekiti State, Nigeria.

### Methodology

A cross-sectional study involving the administration of pretested semi-structured questionnaires to consenting attendees at the Inaugural World NTD day event was conducted on the 30th January, 2020. The questionnaire collected data on attendee's demography, knowledge and awareness about NTDs and its control in Nigeria. Quantitative data were analysed using descriptive statistics in SPSS. 20.0 software and expressed as frequencies and percentages. However, qualitative data to support quantitative analysis were obtained using open-ended questionnaires and analysed thematically.

### Principal findings/conclusion

A total of 309 attendees comprising 167 (54.0%) females, and 142 (46.0%) males participated in this study. By age groupings, majority 206 (66.7%) were within 15–25 years. 167 (54.8%) of the attendees have not heard about NTDs before, whereas 77(35.0%) have

**Data Availability Statement:** All relevant data are within the manuscript and its Supporting Information files.

**Funding:** We express our deep appreciation to Crown Prince Court of Abu Dhabi (www. worldntdday.org) for providing HOM the microgrant for the Inaugural World NTD day event in Ekiti State, Nigeria. The funders had no role in study design, data collection and analysis, decision to publish, or preparation of the manuscript.

**Competing interests:** The authors have declared that no competing interests exist.

heard about NTDs through the advertisement of the event. 181(63.3%) were aware of ongoing NTD control programs in schools and communities. Also, 246 (83.4%) of them have not taken or do not know anyone that has taken drugs donated in schools or communities. The number of attendees 41(13.3%) who incorrectly classified malaria as NTDs is higher than those who recognized onchocerciasis 36 (11.7%) and worm infections 34(11.0%) as NTDs (p>0.05). This study has shown that awareness and knowledge about NTDs control activities in Ekiti State is low, thus justifying the event as an awareness day for addressing NTDs. Public enlightment and regular promotional activities such as media engagement will raise the public appreciation and participation in NTDs control activities.

## Author summary

Neglected Tropical Diseases has remained a public health menace in most developing countries including Nigeria. Efforts targeted at controlling the diseases requires substantial community acceptance and participation in drug donation campaigns. This study, therefore, provides information on public knowledge and awareness about NTDs and its control activities. We surveyed 309 attendees during the inaugural 2020 World NTD Day event in Ekiti, Nigeria. Our results show that attendees' knowledge and awareness about NTD control programme and promotional activities is low. There is thus a need for NTD control program managers to engage in regular promotional activities such as media engagements to ensure improved awareness and community involvement.

## Introduction

Neglected Tropical Diseases (NTDs) represents a diverse group of communicable, disabling, chronic and disfiguring diseases that prevail in tropical and subtropical conditions [1,2,3]. NTDs affect more than one billion people in 149 countries, most especially the rural and disadvantaged urban settings challenged by poverty. These areas are usually characterized by inadequate access to water and sanitation facilities, close proximity to infectious vectors and domestic animals or livestock [4–6]. Based on recent classifications, NTDs comprises of 20 different type of diseases with the addition of snakebite, scabies and other ectoparasites, mycetoma and chromoblastomycosis and other deep mycoses [1–3,7]. The newly launched WHO 2021–2030 NTDs roadmap therefore identifies the specific targets and milestones set for each of the 20 diseases, with 2, 11 and 8 of the diseases targeted for eradication, elimination and control respectively by 2030 [8]

NTDs are more common in sub-Saharan Africa (SSA) with about 500 million people affected [5–7,9]. About 25% of Africa's NTD burden is in Nigeria [3,7,9]. Although the epidemiological profiles of snakebite, scabies and mycetoma remain largely unknown in Nigeria, over half of the WHO recognized NTDs are present in Nigeria [10–13]. The country also has the highest number of people infected or at risk in SSA with NTDs amenable to preventive chemotherapy (PC-NTDs) such as onchocerciasis, schistosomiasis, lymphatic filariasis and soil-transmitted helminths (STH) [3,9,14,15].

The 2012 London Declaration achieved a long-term commitment from pharmaceutical companies to donate essential medicines for the control of PC-NTDs in endemic countries, including Nigeria [16]. Donated medicines are delivered at large scale in a coordinated manner to risk groups in schools and households/communities via already established NTDs control

programme architecture at sub-national/district levels [17]. This strategy known as mass administration of medicines (MAM) requires substantial commitment of both financial and human resources [18], and more importantly, effective communal participation in the bid to sustain progress towards established public health targets or goals [17–19]. However, there are pockets of evidence on the low public awareness about NTDs and efforts targeted at controlling it in Nigeria [14,20–23]. This study therefore aims to investigate public awareness about NTDs and efforts targeted at controlling it in Ekiti State, where such information is lacking.

NTDs were one of the few health and developmental issues that didn't have a dedicated advocacy milestone. The World Health Organization has therefore set aside 30th January of each year as the annual World NTD Day to mobilize greater attention, action and investment on priority issues [24]. The annual World NTD day event therefore serves as an exciting opportunity to investigate the knowledge and perception of the public regarding NTDs and its control activities as it brought together stakeholders living or working in endemic countries and across the diverse NTD landscape. This study was therefore cconducted on 30th of January, 2020 during the Inaugural World NTDs day event held in Ekiti State, Nigeria.

## Methodology

### Ethics statement

This study received ethical approval from the Ethics Review Committees of State Primary Health Care Development Agency, Ekiti State, and Federal University Oye-Ekiti, Ekiti State, Nigeria (HRS/WNTD/001). Those who agreed to participate were given informed consent forms to complete after the purpose of the study has been explained to them. Formal consent was obtained through a duly completed consent form with the name and signature of the attendee.

### Study area

This study was carried out in Ekiti State, Nigeria. Ekiti State is one of the six southwestern states in Nigeria. It has 16 Local Government Areas (LGAs) with Ado-Ekiti as the capital. (Fig 1). The state is endemic for Onchocerciasis, Lymphatic Filariasis, Schistosomiasis and Soil Transmitted Helminthiasis. MAM targeted at controling and possibly eliminating these diseases commenced in Ekiti State in 2015 with support from Mission to save the helpless (MITOSATH), a non-governmental organization.

### Planning and preparatory activities

As part of the inaugural commemoration event for the World NTD Day, a round-table stakeholders meeting was held on Thursday, 30th January 2020 at the Faculty of Science auditorium, Federal University Oye-Ekiti. The event was well-advertised via a rally, live radio programme, and the web (email, Twitter, WhatsApp and Facebook). Invitation to attend was openly announced to all interested members of the public.

### Questionnaire administration

The study was cross-sectional in design and involved the administration of already pretested semi-structured questionnaires to collect both quantitative and qualitative data from consenting participants (S1 Text). Participants were approached at the registration desk of the event and invited to participate in the study. Information such as participants' demographic data, and their knowledge and awareness about NTDs were assessed using the pretested questionnaires. To prevent event-induced responses and biases, attendees completed the questionnaires before proceeding with registration or gaining access to the event hall.

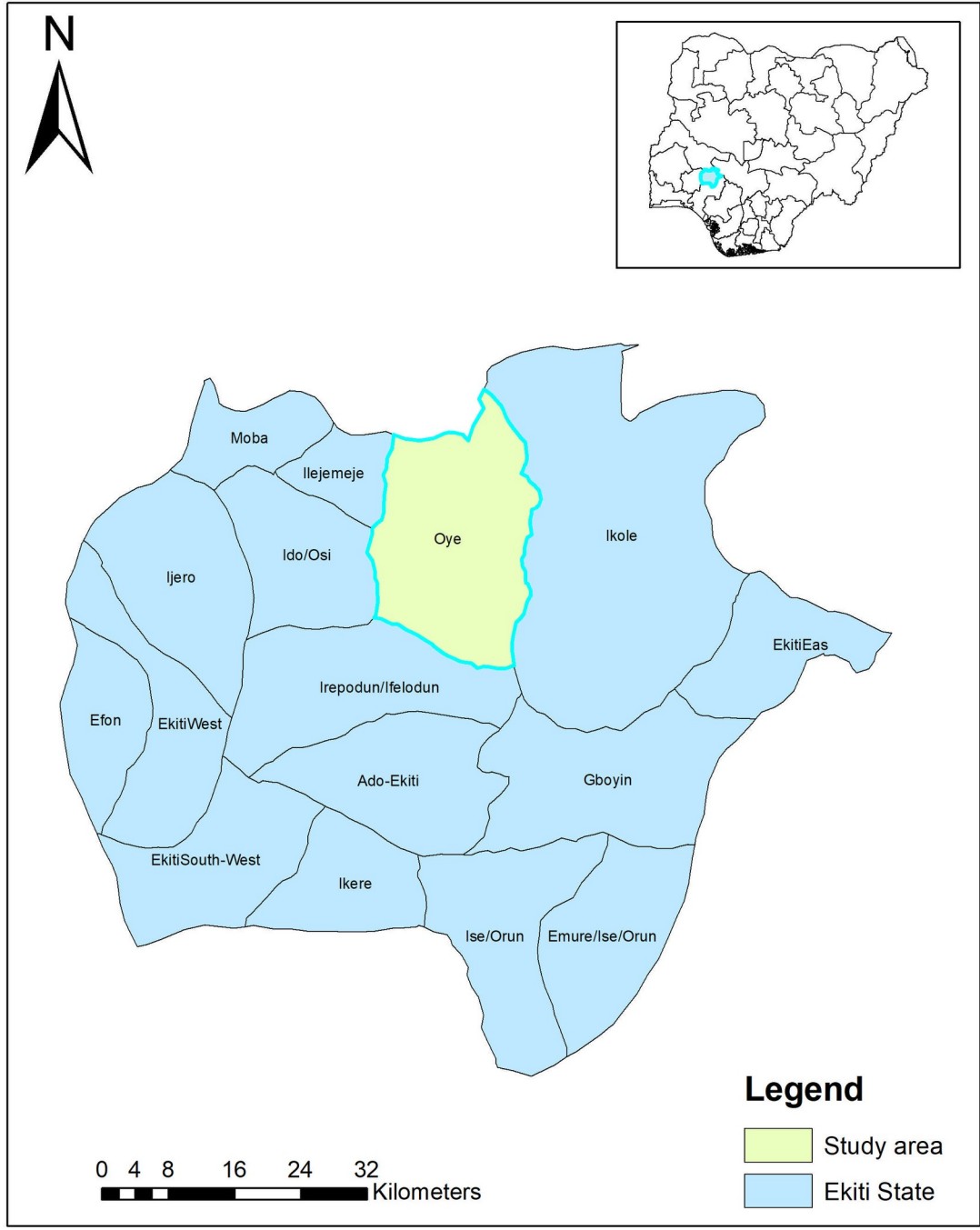

**Fig 1. Map showing the study area with Nigeria as inset.** Map of Fig 1 was created using ArcGIS software by ESRI (www.esri.com). ArcGIS and ArcMap are the intellectual property of Esri and are used herein under license. Copyright Esri. All rights reserved. For more information about Esri software, please visit www.esri.com.

### Data collection, processing, and analysis

Data obtained were entered into Microsoft Excel 2019 software (S1 Data). Quantitative data were analysed using descriptive statistics in SPSS. 20.0 and expressed as frequencies and percentages. Cross-tabulations and Chi-square analysis were performed to investigate

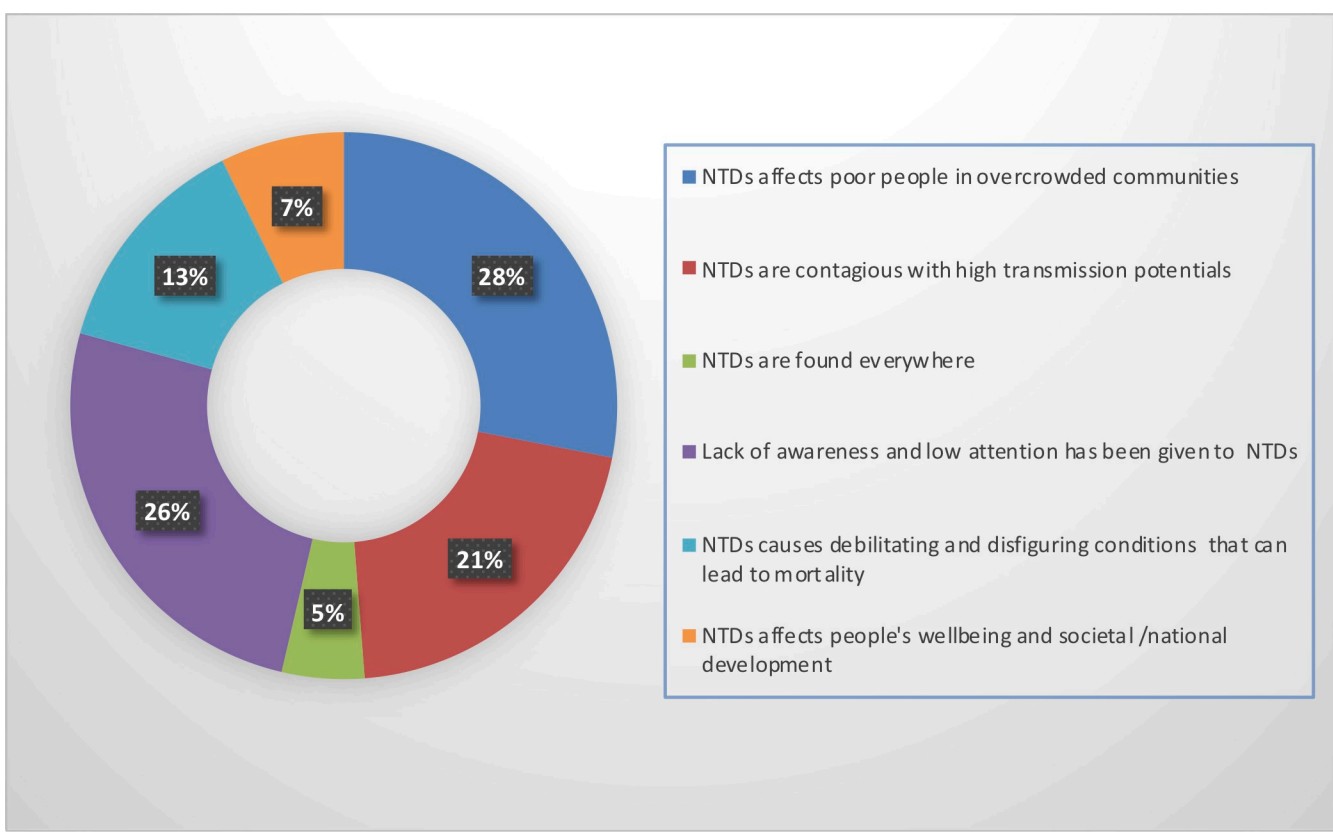

**Fig 2. Perceptions of attendees on why NTDs is a public health problem.**

associations. Confidence interval was set at 95%. However, qualitative data to support quantitative analysis were obtained using open-ended questionnaires, manually coded based on emerging themes and summarized in form of a pie chart (Fig 2). Detailed STROBE checklist for the study has been provided (S1 STROBE Checklist)

## Results

### Demographic information of study participants

A total of 309 attendees participated in the study, including 167 (54.0%) females, and 142 (46.0%) males. Also, the majority of the study participants 206 (66.7%) were within 15–25 years age category, and 4 (1.3%) participants were below 15 years of age. A total of 221 (71.5%) attendees were students or job-seekers, 48 (15.5%) work in tertiary educational institutions, 38 (12.3%) were health workers, 1 (0.3%) was a policy maker and 1 (0.3%) media personnel. (Table 1)

### Perception of attendees about NTDs as a disease and a public health problem

Majority of the attendees, 167 (54.8%) have not heard about NTDs before. The proportion of attendees who have heard about NTDs were not significantly different by occupational status (p = 0.217). Among those who have heard about NTDs, 110 (50.00%) learnt about in the hospital, and 77 (35.0%) through the advert for the World NTD event. Other sources of

**Table 1. Demographic characteristics of study participants.**

|  | Number of respondents | Percentage |
|---|---|---|
| **Sex** |  |  |
| Male | 142 | 46.0 |
| Female | 167 | 54.0 |
| Total | 309 | 100.0 |
| **Age (in years)** |  |  |
| <15 | 4 | 1.3 |
| 15–25 | 206 | 66.7 |
| 26–35 | 39 | 12.6 |
| >35 | 60 | 19.4 |
| Total | 309 | 100.0 |
| **Occupation** |  |  |
| Students/unemployed | 221 | 71.5 |
| Academia* | 48 | 15.5 |
| Health professionals** | 38 | 12.3 |
| Policymakers | 1 | 0.3 |
| Media personnel | 1 | 0.3 |
| Total | 309 | 100.0 |

*Academia was defined as anyone working in a tertiary educational institution including lecturers, researchers and administrative staff

**Health professionals was defined as anyone working in health institutions including hospitals, ministries of health, primary health care centers and is involved in the disease control programme and practice.

information listed were schools 18 (8.2%) and internet 15 (6.8%) (Table 2). There were significant differences in the proportions across the different occupational status (p = 0.004). Also, a total of 87 (28.2%) and 61 (19.7%) attendees were able to correctly identify schistosomiasis and lymphatic filariasis respectively as NTDs. Onchocerciasis and Soil-Transmitted Helminthiasis were also identified as NTDs by 36 (11.7%) and 34(11.0%) additional attendees respectively (Fig 2). Other significant diseases such as Malaria, Tuberculosis, HIV, and Diabetes were incorrectly listed as NTDs by 41 (13.3%), 27 (8.7%), 17 (5.5%) and 14 (4.5%) attendees respectively (Fig 3). Majority of the attendees, 232 (88.5%) also think that NTDs is a public health problem, although the proportions were not significantly different across the occupational status of attendees (p = 0.853). A sub-stratum of the attendees, 82 (26.5%) ascribed several reasons to it, some of which include; NTDs affects people in overcrowded communities (23 (28%)), and the fact that poor awareness and low attention has been given to it (21(26%)) (Fig 2)

## Perception of attendees about signs and symptoms of common NTDs

A total of 166 (55.5%) of the attendees have not seen a probable NTDs case before. There was insignificant difference in the proportion of attendees who have seen a probable case and those who have not across the occupational status of attendees (p = 0.551). Among the 4 major symptoms of PC-NTDs (bloody urine, swollen limbs, worm(s) in the stool and depigmented skin) selected for this study, a high proportion of the attendees 217 (71.9%), 207 (70.2%), 179 (59.9%) and 166 (54.8%), indicated that they have not seen anyone with bloody urine, depigmented skin, wormy stool and swollen limbs before the event respectively. Significant difference only exists among the proportions of attendees who have seen anyone with swollen limbs or thighs by occupational status (p = 0.002) (Table 3)

**Table 2. Knowledge of study participants about NTDs.**

| | | Occupational status of study participants | | | | | | |
|---|---|---|---|---|---|---|---|---|
| | Students/ unemployed | Academia | Government official | Media personnel | Health professionals | Total | $\chi^2$, Df, p-value |
| **Have you heard about NTDs before?** | | | | | | | |
| Yes | 93(42.5) | 27(58.7) | 0(0) | 0(0) | 18(47.4) | 138 (45.2) | 5.764, 4, 0.217 |
| No | 126(57.5) | 19(41.3) | 1(100) | 1(100) | 20(52.6) | 167 (54.8) | |
| Total | 219(71.8) | 46(15.1) | 1(0.3) | 1(0.3) | 38(12.5) | 305(100) | |
| **Where did you learn about it?** | | | | | | | |
| Hospital | 86(53.4) | 17(47.2) | 0(0) | 0(0) | 7(31.8) | 110 (50.0) | 17.070, 9, 0.048 |
| The advert of this event | 57(35.4) | 13(36.1) | 0(0) | 1(100) | 6(27.3) | 77 (35.0) | |
| School | 11(6.8) | 2(5.6) | 0(0) | 0(0) | 5(22.7) | 18 (8.2) | |
| Internet | 7(4.3) | 4(11.1) | 0(0) | 4(18.2) | 15(6.8) | 15 (6.8) | |
| Total | 161(73.2) | 36(16.4) | 0(0) | 1(0.5) | 22(10.0) | 309 (100) | |
| **Do you think NTD is a public health problem?** | | | | | | | |
| Yes | 163(89.1) | 37(84.1) | 1(100) | 1(100) | 30(90.0) | 232 (88.5) | 1.352, 4, 0.853 |
| No | 20(10.0) | 7(15.9) | 0(0) | 0(0) | 3(9.1) | 30 (11.5) | |
| Total | 183(69.8) | 44(16.8) | 1(0.4) | 1(0.4) | 33(12.6) | 262 (100) | |

## Awareness about NTD control programming and promotional activities

A total of 181 (63.3%) of the attendees were aware of ongoing NTD control programmes in schools and communities. There was insignificant difference in the proportion across the occupational status of attendees (p = 0.056). A high proportion 246 (83.4%) of the attendees have not taken or do not know anyone that has taken PC-NTD drugs/medicines donated in schools or communities (Table 4). There was insignificant difference in the proportion across

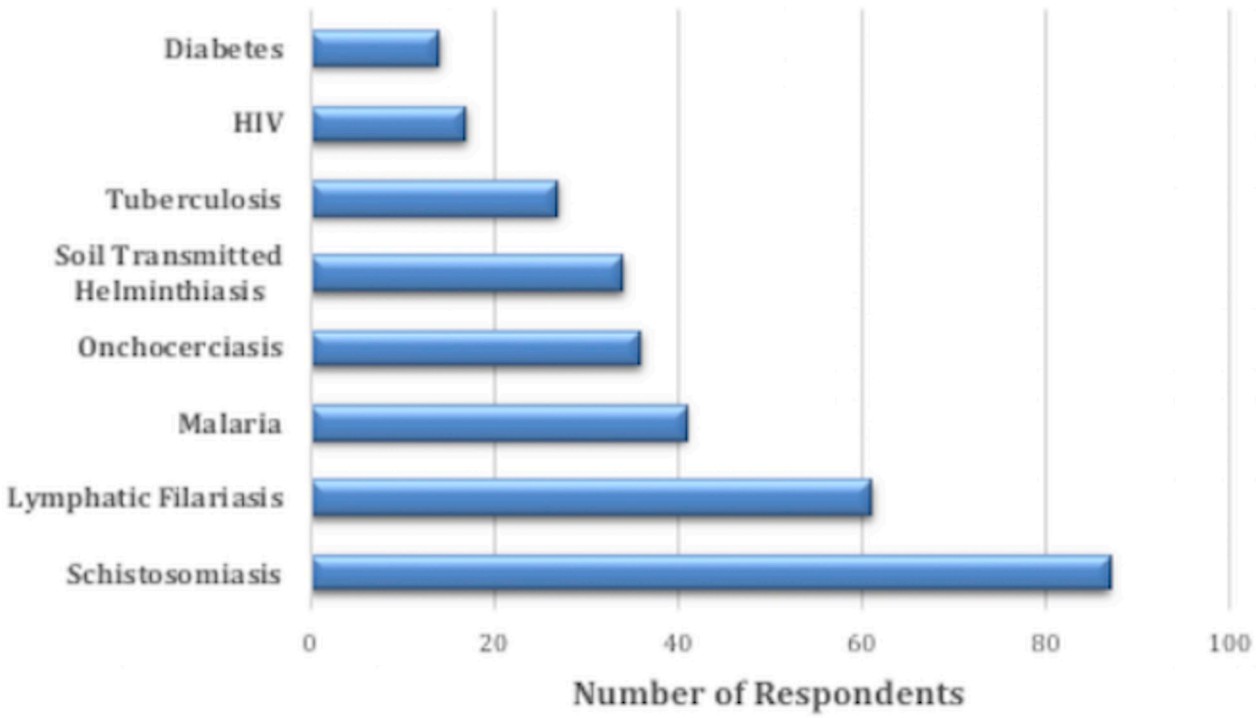

**Fig 3. Perceptions of attendees on what NTDs are.**

**Table 3. Awareness about signs and symptoms of NTDs.**

| | | Occupational status of study participants | | | | | |
|---|---|---|---|---|---|---|---|
| | Students/ unemployed | Academia | Government official | Media personnel | Health professionals | Total | $\chi^2$, Df, p-value |
| **Have you seen anyone affected by NTDs before?** | | | | | | | |
| Yes | 92(43.2) | 24(51.1) | 0(0) | 1(100) | 16(43.2) | 133(44.5) | 3.040, 4, 0.551 |
| No | 121(56.8) | 23(48.9) | 1(100) | 0(0) | 21(56.8) | 166 (55.5) | |
| Total | 213(71.2) | 47(15.7) | 1(0.3) | 1(100) | 37(12.4) | 299 (100) | |
| **Have you seen anyone with swollen limbs or thigh before?** | | | | | | | |
| Yes | 88(40.7) | 33(70.2) | 1(100) | 1(100) | 14(36.8) | 137 (45.2) | 17.101, 4, 0.002 |
| No | 128(59.3) | 14(29.8) | 0(0) | 0(0) | 24(63.2) | 166 (54.8) | |
| Total | 216(71.3) | 47(15.5) | 1(0.3) | 1(0.3) | 38(12.5) | 303 (100) | |
| **Have you seen anyone with bloody urine before?** | | | | | | | |
| Yes | 55(25.7) | 18(37.5) | 0(0) | 0(0) | 12(31.6) | 85 (28.1) | 3.714, 4, 0.446 |
| No | 159(74.3) | 30(62.5) | 1(100) | 1(100) | 26(68.4) | 217 (71.9) | |
| Total | 214(70.9) | 48(15.9) | 1(0.3) | 1(0.3) | 38(12.6) | 302 (100) | |
| **Have you seen anyone with a worm in the stool before?** | | | | | | | |
| Yes | 76(35.7) | 26(55.3) | 0(0) | 0(0) | 18(48.6) | 120 (40.1) | 8.726, 4, 0.068 |
| No | 137(64.3) | 21(44.7) | 1(100) | 1(100) | 19(51.4) | 179 (59.9) | |
| Total | 213(71.2) | 47(15.7) | 1(0.3) | 1(0.3) | 37(12.4) | 299 (100) | |
| **Have you seen anyone with depigmented skin before?** | | | | | | | |
| Yes | 56(26.8) | 19(40.4) | 0(0) | 1(100) | 12(32.4) | 88 (29.8) | 6.338, 4, 0.175 |
| No | 153(73.2) | 28(59.6) | 1(100) | 0(0) | 25(67.6) | 207 (70.2) | |
| Total | 209(70.8) | 47(15.9) | 1(0.3) | 1(0.3) | 37(12.5) | 295 (100) | |

the occupational status of attendees (p = 0.276). Also, majority of the attendees 218 (76.5%) have not received health educational messages on NTDs before. Among the few that had received, the listed sources of information were in schools 26 (8.4%), hospitals 8 (2.6%), on radio programmes 4 (1.3%) and through the internet 2 (0.6%). There was no significant difference in the proportions across the occupational status of the attendees (p = 0.337). Only 5 (8.3%) of the attendees think there is enough awareness about NTDs. In addition, majority of the participants 249 (80.6%) are willing to participate in NTD promotional activities including advocacy and health promotion 118 (38.2%), research 63 (20.4%) and networking 54 (17.5%). (Table 5).

## Discussion

Public awareness and perception about neglected tropical diseases are important factors in policy formulations, programme planning and modification of field practices focused at controlling NTDs [14,25]. The World NTD Day was recognized to call for action towards ending the burden of the NTDs via increasing advocacy for support, acknowledging strides, and increasing overall awareness about these group of debilitating diseases [2]. In this study, we assessed the awareness and perception of attendees at the inaugural World NTD Day event in Ekiti State, Nigeria. Findings showed that almost half of the attendees are unaware of NTDs before the event. This supports the existing notions that public awareness about NTDs is still low in Nigeria [20–23]. Furthermore, knowledge about NTD control programming and promotional activities is very poor, and this finding corroborates with that of [14].

More than 80% of the attendees do not know anyone who has benefitted from preventive chemotherapy mass drug administration and about 70% have not received any NTD health

**Table 4. Awareness about NTDs control programme in schools and communities.**

| | Occupational status of study participants | | | | | Total | $\chi^2$, Df, p-value |
|---|---|---|---|---|---|---|---|
| | Students/ unemployed | Academia | Government official | Media personnel | Health professionals | | |
| **Are you aware of any ongoing treatment programme for NTDs in schools or communities?** | | | | | | | |
| Yes | 139(67.5) | 20(46.5) | 0(0) | 1(100) | 21(60.0) | 181 (63.3) | 9.230, 4, 0.056 |
| No | 67(32.5) | 23(53.5) | 1(100) | 0(0) | 14(40.0) | 105 (36.7) | |
| Total | 206(72.0) | 43(15.0) | 1(0.3) | 1(0.3) | 35(12.2) | 286 (100) | |
| **Do you know anyone that has taken the drug/ medicine before?** | | | | | | | |
| Yes | 29(13.8) | 12(25.5) | 0(0) | 0(0) | 8(22.2) | 49 (16.6) | 5.107, 4, 0.276 |
| No | 181(86.2) | 35(74.5) | 1(100) | 1(100) | 28(77.8) | 246 (83.4) | |
| Total | 210(71.2) | 47(15.9) | 1(0.3) | 1(0.3) | 36(12.2) | 295 (100) | |
| **Do you know the name of the drug/medicine?** | | | | | | | |
| Yes (Mectizan/Albendazole/Praziquantel) | 6(2.7) | 3(6.2) | 0(0) | 0(0) | 3(7.9) | 12(3.9) | 9.435, 8, 0.307 |
| Yes (other drugs/medicine) | 1(0.5) | 2(4.2) | 0(0) | 0(0) | 0(0) | 3(1.0) | |
| No response | 214(96.8) | 43(89.6) | 1(100) | 1(100) | 35(92.1) | 294 (95.1) | |
| Total | 221(71.5) | 48(15.5) | 1(0.3) | 1(0.3) | 38(12.3) | 309 (100) | |

educational benefits prior this event. Ekiti is one of the rural states in Nigeria that has benefitted from the community-directed treatment with Ivermectin (CDTI) programme aimed at eliminating onchocerciasis since 2002 [26], and more recently from MAM since 2015 [16]. It is thus surprising that such a huge proportion of the attendees have not benefitted or heard about PC-NTD control programs. The role of residential proximity to health facilities had been sufficiently discussed by [27], as such our findings might have been influenced by the residential proximity of attendee' to areas where MAM were taking place. MAM are usually implemented in rural communities and quite impactful in endemic areas with less dynamic and diverse populations such as those found in rural areas compared to urban areas that are geographically larger in size and complex in human population structure, making MAM more difficult to implement [28]. However, this study did not capture more specific information about attendee mobility and residential status to provide more explanations to the aforementioned low awareness about PC-NTD control programme.

On the other hand, sensitization and advocacy activities for NTDs are conducted once in a year, and usually very close to the set implementation dates for the annual MAM, with the focus of promoting compliance to medicine administration. This strategy offers a very limited time for intensive sensitization about NTDs [17], as compared to all year-round sensitization activities common to vaccination and bed-nets campaigns [29]. Communal support and awareness are heightened when repeated sensitization with community members, leaders and influential community structures, such as religious institutions, takes place [30]. Nevertheless, data from this study emphasizes the need for increased awareness on NTDs control activities and reiterates the importance of annual World NTD day as a complementary platform for

**Table 5. Perception of attendees about public awareness on NTDs as a disease and other NTDs related activities.**

| | Occupational status of study participants | | | | | Total | $\chi^2$, Df, p-value |
|---|---|---|---|---|---|---|---|
| | Students/ unemployed | Academia | Government official | Media personnel | Health professionals | | |
| **Have you received health educational messages on NTDs before?** | | | | | | | |
| Yes | 49(23.9) | 9(21.4) | 0(0) | 0(0) | 9(25.0) | 67(23.5) | 0.778, 4, 0.941 |
| No | 156(76.1) | 33(78.6) | 1(100) | 1(100) | 27(75.0) | 218 (76.5) | |
| Total | 205(71.9) | 42(14.7) | 1(0.4) | 1(0.4) | 36(12.6) | 285(100) | |
| **If yes, where did you receive such message(s)?** | | | | | | | |
| Community | 3(1.4) | 0(0) | 0(0) | 0(0) | 2(5.3) | 5(1.6) | 30.566, 28, 0.337 |
| Hospital | 4(1.8) | 3(6.2) | 0(0) | 0(0) | 1(2.6) | 8(2.6) | |
| Internet | 1(0.5) | 1(2.1) | 0(0) | 0(0) | 0(0) | 2(0.6) | |
| Ministry of Health | 0(0) | 0(0) | 0(0) | 0(0) | 1(2.6) | 1(0.3) | |
| Radio | 1(0.5) | 2(4.2) | 0(0) | 0(0) | 1(2.6) | 4(1.3) | |
| School | 19(8.6) | 3(6.2) | 0(0) | 0(0) | 4(10.5) | 26(8.4) | |
| No response | 171(77.4) | 39(81.2) | 1(100) | 1(100) | 29(76.3) | 241 (78.0) | |
| Cannot remember | 22(10.0) | 0(0) | 0(0) | 0(0) | 0(0) | 22(7.1) | |
| Total | 221(71.5) | 48(15.5) | 1(0.3) | 1(0.3) | 38(12.3) | 309 (100) | |
| **Generally, do you think there is enough awareness about NTDs and NTDs related activities?** | | | | | | | |
| Yes | 4(12.1) | 0(0) | 0(0) | 0(0) | 1(10.0) | 5(8.3) | 2.202, 2, 0.333 |
| No | 29(87.9) | 17(28.3) | 0(0) | 0(0) | 9(90.0) | 55(91.7) | |
| Total | 33(55.0) | 17(28.3) | 0(0) | 0(0) | 10(16.7) | 60(100) | |
| **Are you willing to participate in any NTD promotional related activities?** | | | | | | | |
| Yes | 181(88.7) | 38(86.4) | 1(100) | 1(100) | 28(77.8) | 249 (87.1) | 3.572, 4, 0.467 |
| No | 23(11.3) | 6(13.6) | 0(0) | 0(0) | 8(22.2) | 37(12.9) | |
| Total | 204(71.3) | 44(15.4) | 1(0.3) | 1(0.3) | 36(12.6) | 286(100) | |
| **Which of the following are you willing to participate in?** | | | | | | | |
| Advocacy and health promotion (AHT) | 85(48.6) | 15(36.6) | 1(100) | 0(0) | 17(58.6) | 118 (47.8) | 43.830, 20, 0.002 |
| Networking | 43(24.6) | 3(7.3) | 0(0) | 1(100) | 7(24.1) | 54(21.9) | |
| Research | 44(25.1) | 14(34.1) | 0(0) | 0(0) | 5(17.2) | 63(25.5) | |
| All of the listed | 3(1.7) | 7(17.1) | 0(0) | 0(0) | 5(17.2) | 63(25.5) | |
| AHT and Networking | 0(0) | 1(2.4) | 0(0) | 0(0) | 0(0) | 1(0.4) | |
| AHT and Research | 0(0) | 1(2.4) | 0(0) | 0(0) | 0(0) | 1(0.4) | |
| Total | | | | | | | |

promoting NTD awareness and control activities. Interestingly, about 25% of the attendees indicated that they have heard about NTDs from the advert of the inaugural event which included a rally and live broadcast on a radio channel. This, therefore, reiterates the potential of open symposiums, online radio programme and rallies as important advocacy strategies in promoting health educational messages and intervention activities to end-users in common and public places such as schools, markets and hospitals [31].

It is not uncommon that an appreciable number of the attendees wrongly identified malaria as a neglected tropical disease compared to onchocerciasis and soil-transmitted helminthiasis.

          

This can be attributed to the fact that there is a common ground of acceptance for the morbidities and burden associated with malaria disease. The incubation period for *Plasmodium spp.*, the causative agent of malaria is between 7 and 30 days which is far shorter than the latency period for other NTDs such as onchocerciasis and soil-transmitted helminthiasis which may take several months to more than a decade [32]. Our findings show that majority of the attendees have not seen anyone with bloody urine, swollen limbs, wormy stool and depigmented skin before. This observation is in line with the findings of [19] and may contribute to the reasons why community members perceive less need for MAM over time. Also, unlike NTD control programmes, sustained malaria control programme have been in existence for quite a longer period, and they have more field presence for interventions delivery such as distribution of insecticide treated bednets and intermittent preventive treatment to endemic regions than there are for NTDs control [33]. These might have contributed to more increased awareness for malaria control than most NTDs [14,34].

The WHO 2021–2030 NTDs roadmap clearly identified with awareness-generation activities aimed at educating and informing endemic communities on behavior changes, MDA scheduling, treatment and care options [8]. Without the public being mobilized to participate in NTD control and promotional activities in Nigeria, it may be increasingly difficult to meet set treatment coverage and achieve the desired objective of the WHO 2021–2030 NTDs roadmap of accelerating control and eventual elimination. This study has thus revealed a low level of public perception of NTDs and their control activities among attendees of this event, this is similar to the perception reported among attendees of a job fair in Abuja, Nigeria [14]. Since sustainable promotional and awareness activities require substantial resources commitment and coordination [14,17,19]. Efforts aimed at improving awareness on NTDs, disease control programs and health promotional activities should therefore be modified on this context to ensure optimal communal participation both in rural and urban areas [35]. NTDs largely affect rural populations and resourced challenged areas where access to wealth and media platforms are limited. As such, promotional activities that has a wider coverage and requires lesser financial commitments such as regular radio programmes and dialogues through community meetings should be explored. However, in urban settings, mass public health media campaigns to improve public perception and contribute towards participation and uptake of NTDs intervention tools should be explored. However, in the face of dire financial resources, promotional activities such as engagement with the media via online/TV interviews and podcasts should be done regularly to inform the public about NTDs.

## Limitations

This study is a rapid assessment of attendees' awareness during the maiden World NTD Day event in Ekiti State. The study has the following limitations; (1) the convenient sampling methodology employed in recruiting study participants contributed to the low sample size and restricted sampling frame, (2) to avoid interfering with attendee's participation in the event, data collectors do not have enough time to engage and probe attendees adequately, which might have led to the non-response rate recorded for some variables in the questionnaire (3) this study did not capture more specific information on attendee mobility and residential status that could provide more explanations to the low awareness about PC-NTD control programme observed.

Based on these limitations, our results are descriptive and cannot be used to make inference on the large populace in Ekiti State. However, this study has highlighted some important concepts on awareness and community engagement in NTD control programming. This study therefore offers useful information and research gaps that should be explored using a more

          

robust sampling methodology. More importantly, there is need to repeat the same knowledge assessment in rural communities using a more systematic sampling approach, utilize focus group discussions and avail the questionnaires in a language that is accessible to rural communities to ensure they can express themselves fully.

## Conclusion

Public knowledge and awareness about NTDs control programme and promotional activities among attendees of the maiden World NTD Day event in Ekiti State, Nigeria is low. Investments in regular radio programmes, community dialogues, and massive public health media campaigns targeted at improving public perception on NTDs and contributing towards uptake of NTDs intervention and participation in NTDs promotional activities should be explored.

## Supporting information

**S1 STROBE Checklist. STROBE Checklist.**
(DOC)

**S1 Text. The questionnaire used for this study.**
(DOCX)

**S1 Data. The raw datasets from the study.**
(XLSX)

## Acknowledgments

We appreciate MITOSATH for donation of additional NTD posters and notebooks for attendees. We also appreciate the study participants and members of staff from the Neglected Tropical Diseases control unit, Ekiti State Primary Health Care Development Agency (SPHDA) for their support during the rally and live radio programme that precede this investigation. Finally, our heartfelt gratitude goes to Mrs Funmilayo S. Oluwafemi, Mrs Tunrayo Oluwadare, and all members of the local organizing committee from Federal University Oye-Ekiti and Afe Babalola University, Ado-Ekiti for their support.

## Author Contributions

**Conceptualization:** Hammed O. Mogaji, Francisca O. Olamiju, Olatunwa J. Olamiju, Uwem F. Ekpo.

**Data curation:** Hammed O. Mogaji, Concilia I. Iyeh, Adedotun A. Bayegun, Olaitan O. Omitola.

**Formal analysis:** Hammed O. Mogaji, Olaitan O. Omitola.

**Investigation:** Hammed O. Mogaji, Ikenna M. Odoh, Concilia I. Iyeh, Abdulhakeem A. Adeniran, Segun I. Oyedeji, Hilary I. Okoh, Cynthia U. Umunnakwe, Uwem F. Ekpo.

**Methodology:** Hammed O. Mogaji, Ikenna M. Odoh, Concilia I. Iyeh, Abdulhakeem A. Adeniran, Segun I. Oyedeji, Hilary I. Okoh, Adedotun A. Bayegun, Olaitan O. Omitola, Cynthia U. Umunnakwe, Francisca O. Olamiju, Olatunwa J. Olamiju, Uwem F. Ekpo.

**Project administration:** Olatunwa J. Olamiju.

**Supervision:** Hammed O. Mogaji, Ikenna M. Odoh, Segun I. Oyedeji, Hilary I. Okoh, Francisca O. Olamiju, Olatunwa J. Olamiju, Uwem F. Ekpo.

**Writing – original draft:** Hammed O. Mogaji, Abdulhakeem A. Adeniran.

**Writing – review & editing:** Hammed O. Mogaji, Ikenna M. Odoh, Concilia I. Iyeh, Abdulhakeem A. Adeniran, Segun I. Oyedeji, Hilary I. Okoh, Adedotun A. Bayegun, Olaitan O. Omitola, Cynthia U. Umunnakwe, Francisca O. Olamiju, Olatunwa J. Olamiju, Uwem F. Ekpo.

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
