## [Decision Letter · Decision Letter 0]

1 Feb 2021

Dear Dr Mogaji,

Thank you very much for submitting your manuscript "Attendee Awareness and Knowledge about Preventive Chemotherapy Neglected Tropical Diseases (PC-NTD) control during the First World Neglected Tropical Diseases Day in Ekiti State, Nigeria." for consideration at PLOS Neglected Tropical Diseases. As with all papers reviewed by the journal, your manuscript was reviewed by members of the editorial board and by several independent reviewers. In light of the reviews (below this email), we would like to invite the resubmission of a significantly-revised version that takes into account the reviewers' comments. 

We cannot make any decision about publication until we have seen the revised manuscript and your response to the reviewers' comments. Your revised manuscript is also likely to be sent to reviewers for further evaluation.

Sincerely,

Antonio Montresor

Associate Editor

Jennifer Keiser

Deputy Editor

Reviewer's Responses to Questions

**Key Review Criteria Required for Acceptance?**

**Methods**

-Are the objectives of the study clearly articulated with a clear testable hypothesis stated?

-Is the study design appropriate to address the stated objectives?

-Is the population clearly described and appropriate for the hypothesis being tested?

-Is the sample size sufficient to ensure adequate power to address the hypothesis being tested?

-Were correct statistical analysis used to support conclusions?

-Are there concerns about ethical or regulatory requirements being met?

Reviewer #1: Yes. However, 

1 - there is a clarification for improvement suggested in one of the stages for analysis that needs to be corrected. 

2 - the thematic categories for the analysis of the qualitative questions should also be included. 

3 - The questionnaire used should be presented as an annex/tabulated summary in this paper

Reviewer #2: The objectives of the study are not well stated

The study protocol is faulty

The population and sample size are not adequate for the study

The statistical analysis are too elementary

**Results**

-Does the analysis presented match the analysis plan?

-Are the results clearly and completely presented?

-Are the figures (Tables, Images) of sufficient quality for clarity?

Reviewer #1: Yes. However, there is a clarification for improvement suggested in one of the stages for analysis that needs to be corrected.

Reviewer #2: The analysis conforms with the results presented

**Conclusions**

-Are the conclusions supported by the data presented?

-Are the limitations of analysis clearly described?

-Do the authors discuss how these data can be helpful to advance our understanding of the topic under study?

-Is public health relevance addressed?

Reviewer #1: No. Conclusion is weak, and public health relevance of this work is not made apparent. Needs to be improved.

Reviewer #2: The conclusion is vague due to inadequate and inappropriate sample size

**Editorial and Data Presentation Modifications?**

Reviewer #1: Data modification needs to incorporate demographic data available on the socio-economic status and residence of the respondents. When this is done, the analysis and data presentation would need to be changed accordingly.

Reviewer #2: (No Response)

**Summary and General Comments**

Reviewer #1: This paper is a welcome contribution to inform a structured dialogue on effective communication to support community engagement in NTD programmes. There are not many publications that have attempted to do this for NTDs programmes. However, this paper's scholarship though is weak, given that the data it is using is obtained from a convenient sample from a single event. The conclusions and suggestions for future work need to be made with this in mind, so as to not overstate what the data obtained in this study depicts as it cannot be reliably used to infer upon the entire population in Ekiti state.

Reviewer #2: The manuscript though interesting has some major drawbacks.

-the objectives of the study are not well defined

-the map or coordinates of the site or state is not shown or given

-the authors did not state how the 16 LGAs of the state were mobilized,

-the poor mobilization accounted for the poor sampling frame as shown by the demographic status of the respondents,where most of them where students.PC-NTDs are mostly targeted at rural communities where the disease burden is mostly found and noticed.Urban setting as reflected in this study is a wrong and biased sampling approach

-the questionnaire is faulty and wrongly framed e.g question 2 in table 2 should be "if yes,where.....table3 (have you seen any one with worm in the school) how does the authors expect the respondents to know except they have been inspecting stools in their communities

most of the questions are too ambiguous

-data analysis is too elementary to warrant a meaningful discussion

PLOS authors have the option to publish the peer review history of their article (what does this mean?). If published, this will include your full peer review and any attached files.

Reviewer #1: No

Reviewer #2: No
---

## [Editor Report · Decision Letter 1]

17 Mar 2021

Dear Dr Mogaji,

We are pleased to inform you that your manuscript 'Attendee Awareness about Preventive Chemotherapy Neglected Tropical Diseases (PC-NTD) control during the First World Neglected Tropical Diseases Day in Ekiti State, Nigeria.' has been provisionally accepted for publication in PLOS Neglected Tropical Diseases.

Best regards,

Antonio Montresor

Associate Editor

Jennifer Keiser

Deputy Editor

---

## [Editor Report · Acceptance letter]

24 Mar 2021

Dear Dr Mogaji,

We are delighted to inform you that your manuscript, "Attendee Awareness about Preventive Chemotherapy Neglected Tropical Diseases (PC-NTD) control during the First World Neglected Tropical Diseases Day in Ekiti State, Nigeria.," has been formally accepted for publication in PLOS Neglected Tropical Diseases.

Best regards,

Shaden Kamhawi

co-Editor-in-Chief

Paul Brindley

co-Editor-in-Chief
